# Electrospinning Highly Concentrated Sodium Alginate Nanofibres without Surfactants by Adding Fluorescent Carbon Dots

**DOI:** 10.3390/nano10030565

**Published:** 2020-03-20

**Authors:** Junke Yu, Zhihui Zhao, Jianxin Sun, Cunzhen Geng, Qingxu Bu, Dawei Wu, Yanzhi Xia

**Affiliations:** 1State Key Laboratory of Bio-Fibers and Eco-Textiles, Shandong Collaborative Innovation Center of Marine Biobased Fibers and Ecological Textiles, Institute of Marine Biobased Materials, Qingdao University, Qingdao 266071, China; 2017021033@qdu.edu.cn (J.Y.); 2018025318@qdu.edu.cn (J.S.); qdugcz@qdu.edu.cn (C.G.); 2017021039@qdu.edu.cn (Q.B.); 2017020825@qdu.edu.cn (D.W.); 2School of Material Science and Engineering, Qingdao University, Qingdao 266071, China; 3School of Chemistry and Chemical Engineering, Qingdao University, Qingdao 266071, China

**Keywords:** sodium alginate, electrospinning, nanofibres

## Abstract

In this study, sodium alginate (SA) nanofibres were obtained by electrospinning via the assistance of traditional poly(ethyl oxide) (PEO) and dimethyl sulfoxide (DMSO) with a high SA/PEO ratio of up to 94:6. However, surfactants with more or less toxicities were replaced by nontoxic and fluorescent carbon dots (CDs) to improve spinnability. Experimental details were conducted by fixing the ratio of SA/PEO to 90:10. Then, the electrospinning products of solutions with different compositions were observed with scanning electron microscopy. Properties such as conductivity, surface tension and rheology of the solutions were investigated to determine the key influencing factors. Moreover, since CDs have excellent fluorescence properties, the fluorescent properties of the nanofibre membrane that was blended with CDs were then collected. In addition, in vitro cytotoxicity assessment of the nanofibres were conducted to evaluate the toxicities and biocompatibility.

## 1. Introduction

Electrospinning has become one of the main methods for the efficient preparation of nanofibrous materials due to its simple manufacturing equipment, low spinning cost, variety of spinnable materials and controllable processes [1,2]. Electrospun nanofibres have the advantages of a large specific surface area, high porosity, adjustable pore size [3,4,5], etc. and have potential applications in drug delivery, wound dressing, biosensing, and other fields of materials science and biology [6].

Sodium alginate (SA) is a natural polysaccharide extracted from brown algae. SA is a linear polymer composed of two glucose units, α-L-mannuronic acid and β-D-guluronic acid, which are randomly linked by 1,4-glycosidic bonds [4,7,8]. SA is nontoxic and has excellent biocompatibility and biodegradability [9,10], thus, it is widely used in bioengineering fields in applications such as tissue scaffolding and drug delivery [11]. However, the electrospinning of SA is very difficult to achieve because negative charges from the abundant carboxyl groups on the SA chains lead to strong electrostatic repulsion of the intra- and inter- SA chains; the above makes it impossible to form an effective chain entanglement to realize electrospinning. Many efforts have been made towards effectively electrospinning SA. Two main approaches have proven effective, namely incorporating a co-solvent and/or a second polymer into the electrospinning system. With respect to the co-solvent, glycerol [11] and ethanol [12] have been used for successful preparation of electrospun SA nanofibres, but this method has not been widely used. The method of adding a second polymer is more widely used because it allows for simple operation and better control. Polymers that can be electrospun from aqueous solutions, such as poly(vinyl alcohol) and poly(ethylene oxide) (PEO) [13], and from water-soluble biopolymers, such as collagen [14] and chitosan [15], are usually used as the second polymer. Among them, low-cost PEO is the most popular second polymer due to its biocompatibility, biodegradability, and nontoxicity. In addition, it forms H-bonds with SA, reducing the viscosity of the solution. Lu et al. prepared SA/PEO blended nanofibres with the SA/PEO ratio of 75:25 [13]. Saquing et al. obtained nanofibres with the SA/PEO ratio of 85:15 by the addition of a Triton X-100 surfactant [16]. Bhattarai et al. obtained nanofibres with 90:10 SA/PEO ratio by adding Triton X-100 and additional dimethyl sulfoxide (DMSO) as a co-solvent [17], but the nanofibres resulted in having a relatively poor size distribution. Later, Kyzioł et al. selected a less toxic nonionic surfactant, Pluronic F-127, to replace the toxic Triton X-100, but the highest SA/PEO ratio was only 80:20 [18]. However, there are still two questions that remain when using this method: (1) how can the concentration of SA in the nanofibres be further increased and (2) is it possible to avoid using the expensive and more or less toxic surfactants?

Here, we present a simple method to obtain SA nanofibres at a high SA/PEO ratio based on previous works. While the other reaction conditions are kept under the optimization conditions of the above research, instead of using surfactants that are more or less toxic, a small amount of fluorescent, biocompatible, metabolizable and nontoxic carbon dots (CDs) [19] are introduced into the spinning solution, which greatly enhances the chain entanglement of SA macromolecules. Continuous nanofibres with an SA/PEO ratio of up to 94:6 are successfully prepared. Meanwhile, the nanofibres inherit the fluorescence property from the CDs, which may have potential applications in the fields of fluorescent biological imaging and authenticity recognition. In vitro cytotoxicity of the nanofibres were tested and the membranes not only had non- toxicity to cells, but also the SA/CDs nanofibres had the effect of promoting cell growth.

## 2. Materials and Methods

### 2.1. Materials

SA (white powder, Mw = 220 kDa, MWD = 1.5 and M/G = 1.05, which we used before in [20]) was purchased from Qingdao Gather Great Ocean Algae Group (Qingdao, China). PEO (Mw = 1000 kDa) was purchased from Aladdin Industrial Corporation (Shanghai, China). Triton X-100 was bought from Sinopharm Chemical Reagent Co., Ltd (Shanghai, China). Other chemicals were supplied by Sinopharm Chemical Reagent Co., Ltd (Shanghai, China). CDs with a concentration of 60 mg/mL were prepared by a hydrothermal reaction of citric acid and ethanediamine, according to Zhu’s report [21]. MTT (3-[4,5-dimethyl-2-thiazolyl]-2,5-diphenyl-2H-tetrazolium bromide) was purchased from China Center for Type Culture Collection (Wuhan, China). Deionized water was used throughout the experiment.

### 2.2. Preparation of Spinning Solution

SA and PEO aqueous solutions at the same concentration of 3 wt.% were prepared and then mixed with weights of 18 g and 2 g, respectively. Then, 2 g of DMSO was added as a co-solvent. Next, 0.2 g Triton X-100 (marked as sample I) or 0.2 g CD solution (marked as sample III) was added and mixed in the above solution. The solutions were then stirred for 6 h at room temperature and stored overnight to remove bubbles before being used as spinning solutions. A spinning solution was prepared for comparison with neither Triton X-100 nor CDs (marked as sample II). Finally, a sample with 18.8 g SA, 1.2 g PEO, 2 g DMSO and 0.2 g CD solution was also prepared (marked as sample IV) to show the limit of the method. Table 1 listed the detailed compositions of the four samples.

### 2.3. Electrospinning Process

The spinning solution was placed in a 10 mL syringe with a 12# metal needle placed at the top. The positive lead of the high voltage power supply was connected to the outer surface of the needle by an alligator clip, and a piece of rectangular (20 × 15 cm) aluminium foil was connected to the ground by wires as a static-collecting substrate. Other conditions were fixed as follows: the distance between the needle and the collecting substrate was 20 cm; the flow rate of the spinning solution was 0.5 mL/h; the voltage was 20 kV; the temperature was 35~40 °C; and the humidity was kept below 30 %RH.

### 2.4. Characterization

The morphology of the nanofibres was observed with scanning electron microscopy (SEM, QUANTA FEG 250, Thermo Fisher Scientific, Massachusetts, MA, USA). Surface tension was tested using an optical contact angle meter (Theta Lite, Biolin Scientific, Gothenburg, Sweden). Solution conductivity was tested by a conductivity meter (Ray-DDS-307, Shanghai Yidian Scientific Instrument Co., Ltd., Shanghai, China). Rheological analysis was performed with a rotational rheometer (PhysicaMCR301, Anton Paar, Graz, Austria). Fourier transform infrared (FT-IR) spectroscopy was recorded with a Nicolet iS50 FT-IR spectrometer (Thermo Fisher Scientific, Massachusetts, MA, USA). Fluorescence spectra of the CD-incorporated nanofibres were measured with a fluorescence spectrophotometer (FluoroMax-4, HORIBA Scientific Jobin Yvon, Paris, France). Confocal laser scanning microscopy (CLSM) images were collected with a Multiphoton Laser Confocal Microscope (Nikon A1R MP, Nikon Corporation, Tokyo, Japan).

### 2.5. In Vitro Cytotoxicity

The cell cytotoxicity of nanofibres prepared with solutions I and III were tested by the MTT assay with L-929 fibroblasts. The cells were first cultured to logarithmic growth stage in DMEM medium supplemented with 10% FBS. After digestion with trypsin-EDTA, the cells were transferred to wells of 96-well plate with the concentration of 1× 10^5^ cells per well and then incubated for 24 h at 37 °C with 5% CO_2_ until a semi-confluent monolayer was formed. The nanofibres dissolved in PBS (Ph = 7.4) were then added to each well with certain amounts to make sure the final concentration was 0, 16, 32, 61, 125, 250 μg/mL, respectively. After that, the cells were further incubated for 24 h. Then, 20 μL MTT with the concentration of 5 mg/mL was added to each well and incubated for another 4 h. Finally, after carefully removing the surfactant, 150 μL DMSO was added to each well and the well plate was shaken in the dark for 10 min to dissolve the crystals. Absorbance of the solutions at 570 nm and 630 nm were measured using a microplate reader (EnSpire, PerkinElmer Inc. Massachusetts, MA, USA) and the cell viability was the ratio of the absorbance of cells cultured with nanofibres and that of the control group. All samples were performed five times to obtain an average value.

## 3. Results and Discussion

### 3.1. Morphology of Nanofibres

For the SA/PEO/DMSO/surfactant system, previous studies obtained nanofibres with the SA/PEO ratio of 90:10 as the top, but the as-spun nanofibres showed a relatively poor distribution of fibre diameters [17]. SA nanofibres that can be continuously spun with perfect cylindrical morphologies, a narrow distribution of diameters, and no beads or spindles are usually prepared with low SA/PEO ratios, showing a maximum of approximately 80:20. Here, by applying a high voltage and keeping the air humidity low, nanofibres with 90:10 SA/PEO ratio were prepared with a good morphology and distribution (Figure 1a). The nanofibres showed perfect cylindrical morphologies with a relatively narrow distribution, with most nanofibre diameters ranging from 142-356 nm. No beads, spindles, or stranded filaments were observed in the image. Electrospinning without surfactants was then attempted, and no nanofibres were observed, as shown in Figure 1b, and only beads or droplets with diverse sizes strung together by filaments were collected. However, without the use of any surfactants, perfect nanofibres with similar morphologies, diameters, and distributions (142–320 nm) were obtained with an addition of a small number of CDs (Figure 1c). Based on this result, we tried to further enhance the SA/PEO ratio in the system, and nanofibres with 94:6 SA/PEO ratio could be successfully achieved (Figure 1d), with morphologies similar to those shown in Figure 1a,b, ranging from 180–420 nm, with occasional observations of a few beads. However, a further attempt to increase the SA/PEO ratio to 95:5 failed, indicating that 94:6 might be the maximum SA/PEO ratio for our method.

### 3.2. Properties of the Spinning Solutions

To understand the effect of CDs in electrospinning, the electrical conductivity, surface tension and rheology of samples I, II and III were investigated. As shown in Figure 2a, the conductivities of the three spinning solutions (I, II, and III) were equivalent, which were 3.38 × 10^4^ μS/cm, 3.26 × 10^4^ μS/cm, and 3.49 × 10^4^ μS/cm, respectively. Since Triton X-100 is a well-known nonionic surfactant and the CDs we used here are reported to be capped with uncharged -OH, epoxy, C=O, C=N, N-H groups, etc., their addition did not have much effect on the conductivity of the solution. This result also indicated that the conductivity of the solution was not a key factor for spinning.

The surface tension of spinning solutions is considered to be an essential influencing factor, and high surface tension usually needs to be decreased by the addition of surfactants to suppress the formation of beads or spindles during SA spinning [18,22]. Here, without the addition of Triton X-100, the surface tensions of samples II and III both increased sharply from 35.69 mN/m to 60.68 mN/m and 57.42 mN/m, respectively. The addition of CDs did not reduce the surface tension that much, and the surface tension of sample III was approximately 1.6 times that of sample I. However, despite the large difference in surface tensions between samples I and III, a perfect nanofibre structure, similar to sample I, was formed after spinning, as shown in Figure 1a,c; the above results indicated that other factors rather than just the surface tension were the controlling factors during spinning. This phenomenon was quite different from the previous reports, in which surface tension was thought to be the greatest effect on the eletrospinnability and none SA nanofibres without beads or spindles had been prepared before under such a high surface tension [9,14,20].

The rheological properties of the three solutions were then investigated, and as shown in Figure 2c,d, the addition of Triton X-100/CDs greatly changed the rheology of the SA/PEO/DMSO solutions. First, the dynamic rheology of the three solutions all showed similar rules: the loss moduli (G′′) were larger than the storage moduli (G′) in the low frequency region and became lower than G′ in the high frequency region, which meant that the solutions changed gradually from a viscous fluid to elastomers with increasing shearing frequency. It should be noted that the viscoelastic transition points, i.e., the intersection of G′ and G′′, of the three solutions occurred in very different regions, and especially that of sample II, which appeared in a much larger frequency region than that of samples I and III. It is known that a low intersection of G′ and G′′ is usually accompanied by a high quality of electrospinning [11]. This is because a low intersection means a decrease in chain repulsion, an increase in chain entanglement, which can create new physical crosslinks and possible networks and are all favourable factors for electrospinning. The increase in chain entanglement and physical crosslinking could be confirmed by the increase in solution viscosity, as shown in Figure 2d. Sample III showed the largest viscosity, sample I the second, and sample II showed the lowest viscosity. Here, at a fixed ratio of SA/PEO/DMSO, the addition of Triton X-100 and CDs both improved the viscoelastic transition of the solution, but the improvement from CDs was greater than that of Triton X-100. This benefited from the numerous functional groups on the surface of the CDs, including -OH, epoxy, C=O, N-H, etc., as mentioned above. These functional groups interacted with the -COOH and -OH groups on the SA chains to form hydrogen bonds and, as a result, disrupted the inter- and intra-electrostatic repulsions of the SA chains, which increased chain entanglement. In addition, the newly formed hydrogen bonds could also serve as physical crosslinking points. Apparently, the increase in chain entanglement and physical crosslinking of sample III was the key factor for the improved spinnability. The influence of Triton X-100 on the rheology of the solution was quite different from our perception, which was thought to improve the spinnability by reducing surface tension rather than influencing the rheology. However, previous studies on the influence of Triton X-100 were usually performed with a simpler SA/PEO system, not the SA/PEO/DMSO system used here; furthermore, the concentrations and ratios of the polymers were also different. The influence of TritonX-100 on the rheology of sample I is under further investigation. All in all, the rheology results of three spinning solutions show that the degree of chain entanglement in solutions I and III was much larger than that in solution II. Combine the results that both the spinnable I and III had similar chain entanglements but rather different surface tensions, we deduce that rheology properties especially degree of chain entanglement rather than surface tension dominated the successful electrospinning of samples I and III. The failure of electrospinning of sample II with smaller chain entanglement degree also confirm our deduction.

### 3.3. Fluorescence (FL) Properties and Structure of Nanofibres

FT-IR spectra of samples I and III were collected and are shown in Figure 3a to clarify the composition of the nanofibres and the possible influence of CDs on the structure of the nanofibres. Since the amount of Triton X-100 and CDs was too low to be distinguished by FT-IR, the two spectra were very similar, and both spectra showed typical peaks of SA/PEO blended nanofibres, as marked in the figure. It should be noted that due to the small amount of PEO in the nanofibres, both the stretching and rocking peaks of -CH_2_- appeared at very low intensity.

Fluorescence properties, especially excitation-dependent-emission (EDE) fluorescence, are a very important advantage of CDs. To determine whether the fluorescence property of CDs was preserved by the nanofibres, fluorescent spectra of sample III were collected. As shown in Figure 3b, the optimal excitation and emission of the nanofibres appeared at 343 nm and 430 nm, respectively, and both slightly blue-shifted from the 360 nm and 443 nm of the original CDs [21] because the change in the surface state of the CDs when incorporated into the nanofibres affected the band gap. The emission barely changed when excited from 320 nm to 350 nm but red-shifted to 495 nm when excited by 405 nm, showing typical EDE fluorescence. EDE could be further verified by the CLSM images. As shown in Figure 3c, straight, homogeneous, and evenly distributed nanofibres could be clearly observed. When exposed to light of 405, 488 and 561 nm, bright and homogenous emission with blue, green, and red colours were observed, respectively. It should be noticed that the azure emission colour of Figure 3d exactly coincided with the maximum emission wavelength of 495 nm in Figure 3b after both exposed to the 405 nm. Moreover, the uniform emission indicated that the CDs were uniformly dispersed in the nanofibres. The bright and EDE fluorescence make the nanofibres good candidates for anti-counterfeiting, bioimaging, medical diagnosis, and so on.

### 3.4. Cytotoxicity and Biocompatibility of Nanofibres

To determine the application potential of in the field of biomedicine, the cytotoxicity and biocompatibility of SA nanofibres were studied. In vitro cytotoxicity assessment of SA nanofibres (sample I) and SA/CDs nanofibres (sample III) were evaluated by using MTT analysis with L-929 fibroblasts. As shown in Figure 4, for sample I, the cell viability was over 99% at the low concentration of 16 μg/mL, but gradually decreased when the concentration increased and kept steady at about 70%, revealing relatively low toxicity towards the cells. However, for sample III, the cell viabilities were over 100% compared to the control group at all concentrations and it increased first and then decreased with the increase of concentration, and arrived the maximum of 144% at 61 μg/mL. The results showed that SA/CDs nanofibres were not only non-toxic and well biocompatible, but also could promote the cell proliferation, making them good candidates to be used as wound dressing. Based on the different cell viabilities of sample I and III, the cell proliferation effect was deduced to be benefited from CDs, which had abundant active sites provided by the amino and hydroxyl groups, through which other biocompatible components such as proteins, cell growth factors, or peptides could be further immobilized [23].

## 4. Conclusions

In this article, we presented a facile method for the preparation of SA nanofibres with a high SA/PEO ratio of 94:6 by electrospinning without surfactants. Nontoxic and fluorescent CDs were used instead of more or less toxic surfactants to improve the electrospinnability of the solution. As CDs offered to have improved the shear viscosity, corresponding surface tension had no effect on electrospinnability confirming that rheological properties played a vital role in successful electrospinning. It was deduced that the improvement of electrospinnability relied on increasing the chain entanglement and physical crosslinking by the interactions between the CDs and SA. Moreover, the fluorescence properties of the CDs were inherited by the nanofibre membrane. The membrane emitted bright light with different colours upon different excitations, which could potentially be used in anti-counterfeiting, bioimaging, medical diagnosis fields and so on. In addition, SA/CDs nanofibres have non-toxicity and good biocompatibility, can promote cell proliferation, and are good candidates for wound dressing.

## Figures and Tables

**Figure 1 nanomaterials-10-00565-f001:**
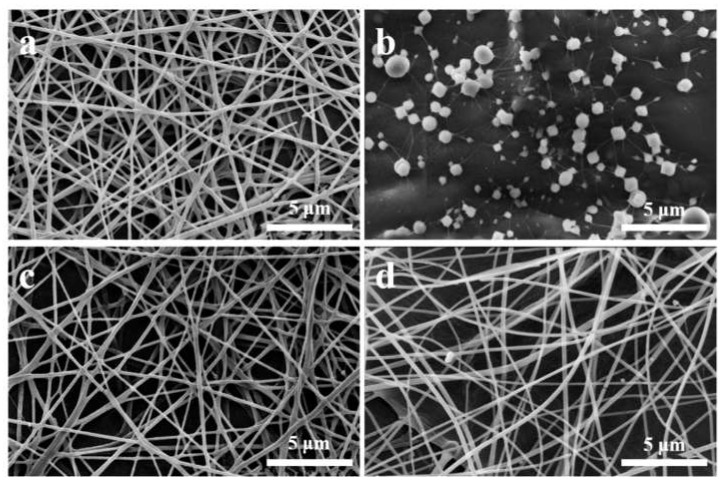
SEM images of SA nanofibres spun with the four different sample solutions: (**a**) I; (**b**) II; (**c**) III and (**d**) IV.

**Figure 2 nanomaterials-10-00565-f002:**
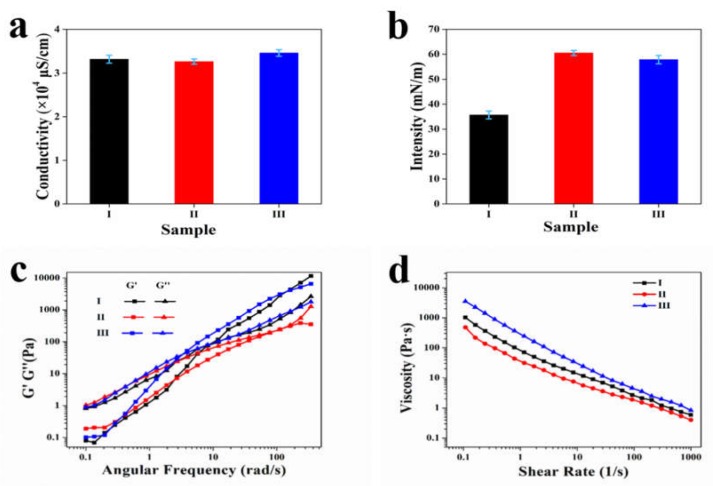
Properties of the four solutions: (**a**) conductivity; (**b**) surface tension; (**c**) storage (G′) and loss (G′′) moduli as functions of frequency and (**d**) viscosity changes with shear rate.

**Figure 3 nanomaterials-10-00565-f003:**
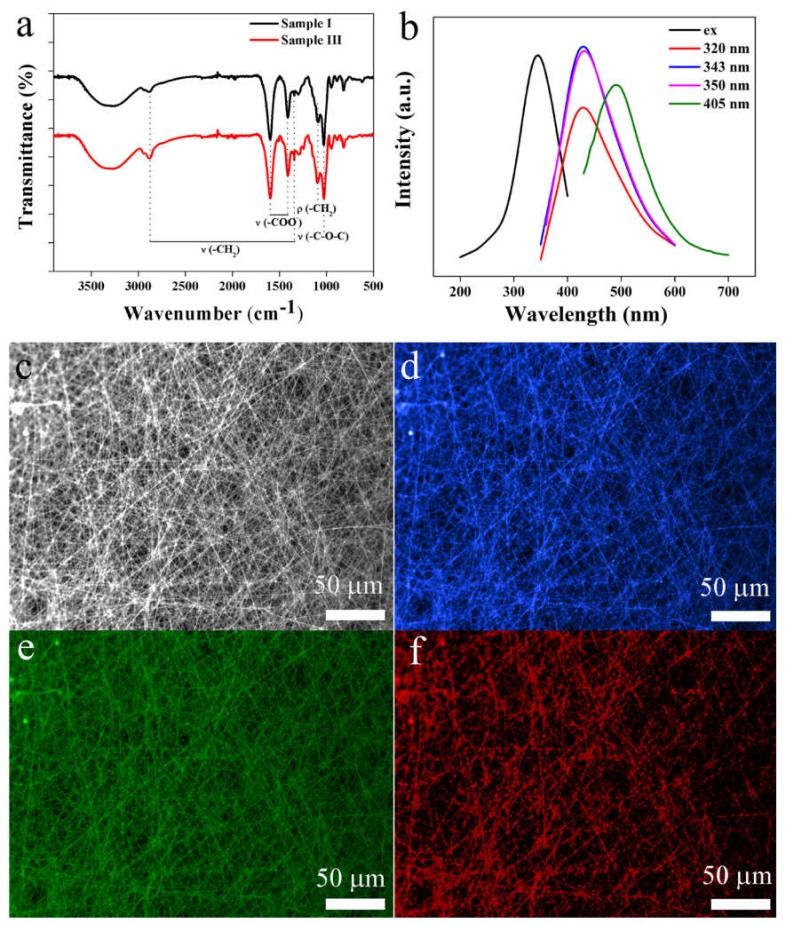
(**a**) FT-IR spectra of samples I and III; (**b**) Fluorescent spectra of sample III, including excitation and emission spectra excited at 320 nm, 343 nm, 350 nm and 405 nm; **c**–**f**: CLSM images of sample III with (**c**) no excitation, and at (**d**) 405 nm, (**e**) 488 nm, and (**f**) 561 nm, respectively.

**Figure 4 nanomaterials-10-00565-f004:**
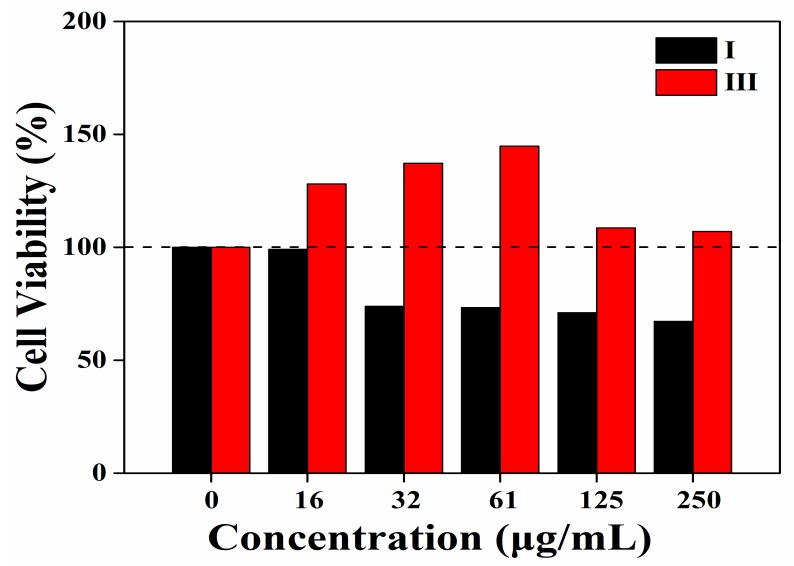
The cell viabilities of L-929 fibroblasts cultured in medium with the nanofibres of different concentrations.

**Table 1 nanomaterials-10-00565-t001:** The compositions of the spinning solutions.

Sample	SA ^a^ (g)	PEO ^a^ (g)	DMSO (g)	Triton X-100 (g)	CDs (g)	SA/PEO Mass Ratio
I	18	2	2	0.2	0	90:10
II	18	2	2	0	0	90:10
III	18	2	2	0	0.2	90:10
IV	18.8	1.2	2	0	0.2	94:6

the concentration of the solution is 3 wt.%.

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
