# Peer review of "Electrospinning Highly Concentrated Sodium Alginate Nanofibres without Surfactants by Adding Fluorescent Carbon Dots"

_nanomaterials, 2020, doi:10.3390/nano10030565_

Round 1

Reviewer 1 Report

Dear Editor,

I have read the manuscript “Electrospinning highly concentrated sodium alginate nanofibres without surfactants by adding fluorescent carbon dots”, and now I am sending my suggestion. The paper focused on the electrospun nanofiber formation of sodium alginate adding PEO, surfactant, and carbon fluorescent dots, in particular, the effect of the carbon fluorescent dots on the nanofiber formation. I think it would be better to be accepted after majorly revised.

  • There were many careless mistakes. For examples, many Chinese characters were remained, and the unit of the electric conductivity was not “µs/cm” but “µS/cm.” In abstract, the word “CD” was used without explanation. I strongly recommend the authors to check the manuscript quite carefully.
  • I could not agree the discussion of the rheological results. The authors wrote the viscoelastic transition point of the sample III was in very different region from that of the samples I and II. But, in the experiment data, the transition of II could not be observed, and the transition points of I and III was observed in the same order of magnitude (a few rad/s). So, the following discussion was not based on the experimental data.
  • In the fluorescent spectra of Fig. 3b, I think there were some mistakes. I think either red or blue curve corresponds to the emission excited at 350 nm. Furthermore, why the authors drew only the excitation spectrum of 320 nm? From these spectra, many misunderstandings would originate.
  • In Fig. 3b, different colors of emission were emitted at 343 and 430 nm. But, the CLSM images in Fig. 3d-f were excited at 405, 488, and 561 nm. I wonder these images corresponded to the identical emission, so I could not find the meaning of three images with different colors.

Reviewer 2 Report

Authors have presented a simple, yet effective approach of employing non-toxic fluorescent carbon dots (CD) instead of conventional toxic surfactants for fabricating electrospun nanofibers of higher sodium alginate concentration. Replacing Triton-X with CDs has shown to have retained the desired morphological and rheological properties of electrospun nanofibers. Moreover, CDs have also rendered cytotoxicity and biocompatibility properties which renders scope for cell proliferation and applications in wound dressing.

General Comments

  1. Section 2.2: Preparation of spinning solution

As this section enables the readers to understand the initial concentration of SA, polymers, additives and co-solvent in solution, it is important to clearly specify the concentration of each constituents of the samples in terms of wt.%. It is rather unclear of why 3 wt.% solutions of SA and PEO (assuming those are dissolved in deionised water) are initially prepared and on what basis it is related to final electrospun nanofibers containing 93 wt. % of SA.

There is still some vagueness in sample description: For instance, sample IV constitutes 18.8 g SA and 1.2 g PEO whereas sample I constitutes 18 and 2 g of SA and PEO respectively. Does sample IV correspond to electrospun nanofibers with highest SA concentration of 94 wt.%

Please revise this section with detailed information on concentration of each material (in wt.%) used for preparing various electrospinning solution (including reference solution). Perhaps, representing the sample description (I, II, III and IV) as separate Table could bring more clarity. Include relevant references if this procedure was adopted from any previous work of authors or others.

  1. Section 3.2. Properties of the spinning solutions

If samples are still available with the authors, characterisation of sample IV ranging from conductivity, surface tension and viscosity would help to support the reported evidences mentioned for sample III.

Specific comments

  • Section 2.1 Materials: Include source for Triton-X
  • Any information to support uniform/complete dispersion of CDs in spinning solution containing SA and PEO? For instance, please include if any ultrasonication steps were done to ensure homogeneity of spinning solutions
  • Page 3, line 95-96: Please check units of relative humidity.
  • Page 5, Figure 2: Please check, there is a mismatch between Figure captions and its respective graphs.
  • Page 5, line 160-164: Despite no considerable increase in surface tension between sample II and III, greater visco-elastic factors (higher shear viscosity) of sample III could have counteracted with correspondingly higher surface tension. As surface tension is one of the critical parameters governing electrospinning, it would be helpful to include such information with the evidence of relevant literature.
  • In that case, line 241-242 of Section 4 must also be revised: ‘As CDs offered to have improved the shear viscosity, corresponding surface tension had no effect on electrospinnability confirming that rheological properties played a vital role in successful electrospinning’

Round 2

Reviewer 1 Report

I checked the revision against my comments, and all points have been revised. I think the paper would be better to be accepted.

Author Response

Dear professor,
Thank you very much for your valuable comments on our work. In the future research, we will definitely work harder. Thanks again for your guidance.